# Sequential Membrane Filtration to Recover Polyphenols and Organic Acids from Red Wine Lees: The Antioxidant Properties of the Spray-Dried Concentrate

**DOI:** 10.3390/membranes12040353

**Published:** 2022-03-23

**Authors:** Polychronis Filippou, Soultana T. Mitrouli, Patroklos Vareltzis

**Affiliations:** 1Laboratory of Food and Agricultural Industries Technologies, Chemical Engineering Department, Aristotle University of Thessaloniki, GR541 24 Thessaloniki, Greece; fpolixronis@hotmail.com; 2Laboratory of Natural Resources and Renewable Energies, Chemical Process and Energy Resources Institute (CPERI), Centre for Research and Technology-Hellas (CERTH), 6th km Charilaou-Thermi Road, GR570 01 Thessaloniki, Greece; tania.mitrouli78@gmail.com

**Keywords:** membranes, filtration, polyphenols, organic acids, spray-drying, oxidation

## Abstract

The vinification process produces a considerable amount of waste. Wine lees are the second most generated byproduct, representing around 14% of total vinification wastes. They are a valuable source of natural antioxidants, mainly polyphenols, as well as organic acids, such as tartaric acid. This paper deals with the application of an integrated, environment friendly membrane separation process to recover polyphenols and organic acids. A two-step membrane process is described, consisting of an ultra- and a nano-filtration process. The physicochemical and antioxidant properties of all the process streams were determined. High Pressure Liquid Chromatography (HPLC) was employed for identifying certain individual organic acids and polyphenols, while the antioxidant potential was determined by the 2,2′-diphenyl-1-picrylhydrazyl radical) (DPPH) radical scavenging ability and ferric reducing ability. A liquid concentrate stream containing 1351 ppm of polyphenols was produced and then spray dried. The resulting powder retained most of the polyphenols and antioxidant properties and was successfully applied to a real food system to retard lipid oxidation, followed by Thiobarbituric Acid Reactive Substances (TBARS) and the determination of oxymyoglobin content. The results show that membrane separation technology is an attractive alternative process for recovering value-added ingredients from wine lees.

## 1. Introduction

Red wine has been widely investigated for its constituents of biological interest, including flavonoid compounds (flavanols, monomeric catechins, proanthocyanidins, anthocyanins, anthocyanidins) and nonflavonoid phenolic compounds (resveratrol), as well as their metabolites [1]. These bioactive compounds of red wine, when consumed moderately, have been linked to a reduced risk of heart disease and extended lifespan [1,2,3].

The vinification process, however, produces considerable amounts of waste. The worldwide production of grapes in 2019 was about 84 mt [4]. It has been estimated that 0.13 t of marc, 0.06 t of lees, 0.03 t of 37 bunches, and 1.65 m^3^ of wastewater are generated for each ton of processed grapes [5]. In fact, grape pomace comprises the highest amount of waste generated in the modern wine industry, representing nearby 62%, while the lees, stalks, and dewatered sludge represent around 14%, 12%, and 12%, respectively [6]. Wine lees, as the second most generated byproduct of vinification process, have not gained proper attention as a potential source of bio-active and value-added components [7]. They are a valuable source of natural antioxidant, and organic acids, mainly tartaric acid [8,9,10]. Even rarer in the literature are reports regarding efforts to produce, test, and characterize dried extracts, which are recognized for their advantages in terms of the higher stability of the active substances over time and the lower storage costs [11,12,13,14].

Currently, phenolic compounds are recovered by extraction with organic solvents such as methanol, ethanol, and acetone [15], which are toxic and/or irritants [16]. Ultrasound- and microwave-assisted extraction have been the focus of research activity in the effort to develop environmentally friendly processes using nontoxic and food-grade solvents [17]. An attractive alternative process is membrane technology for separating, purifying, and concentrating bioactive phenolic compounds from aqueous streams [18]. There are not many studies dealing with the sequential ultra- and nano-filtration of wine lees in particular. Few of them result in a final powdered product, which is evaluated in real food applications. Giacobbo et al. developed a process of aqueous extraction associated with microfiltration for the recovery of natural antioxidants from wine lees generated in the second racking of red winemaking [19]. The main constituents of the permeate were polyphenols, polysaccharides, organic acids, and minerals. The permeate can be further processed by a nanofiltration step to produce a concentrated solution, rich in polyphenols, with high antioxidant activity [20]. Recently, it was shown that nanofiltration membranes can be used in the valorization of wine lees [21]. In another study, the results showed that a nanofiltration membrane with a typical pore size of approximately 1 kDa exhibited satisfactory separation and low-fouling filtration performance when applied to wine lees processing streams [22].

The aim of this study is to further evaluate the sequential application of ultra- and nanofiltration membranes of tubular geometry to separate and concentrate polyphenols from undiluted red wine lees in a cost-effective and environmentally friendly manner. In this study, an integrated process from wastes (wine lees) to a value-added final product (spray-dried antioxidant concentrate) is presented. Individual organic and phenolic acids were identified in the process streams of both membranes. Process streams were characterized for their in vitro anti-oxidant potential, and the concentrated polyphenols liquid stream was dried via spray-drying and tested in a real food system for retarding lipid oxidation.

## 2. Materials and Methods

### 2.1. Materials

Wine lees, produced during the vinification of red Syrah variety grapes were provided by a winery (Mesenikola, Karditsa, Greece) during the 2018–2019 season. The samples were collected from the bottom of a stainless-steel wine stabilization tank. The collected wine lees were transferred in insulated containers filled with ice to the laboratory and stored in a freezer (−20 °C). One day before filtration experiments, frozen samples were placed into a refrigerator to thaw. Then, they were centrifuged at 8000× *g* for 15 min at 4 °C and the supernatant was used as the feed stream for the ultrafiltration step. The physicochemical properties of the wine lees were determined by analyzing them in triplicate (Table 1).

Acetic acid, acetonitrile, methanol, epicatechin, quercetin, ferulic acid, caffeic acid, p-coumaric acid, resveratrol, gallic acid, DL-malic acid, citric acid, Folin–Ciocalteu reagent, sodium carbonate, 2,2-Diphenyl-1-picrylhydrazyl radical (DPPH), ferric chloride, butylated hydroxytoluene (BHT), hydrochloric acid (HCl), trichloroacetic acid (TCA), potassium phosphate, sodium phosphate, 2,4,6-Tris(2-pyridyl)-s-triazine (TPTZ), thiobarbituric acid (TBA), tetraethoxypropane, and sodium hydroxide (NaOH) were purchased from Sigma Aldrich (Merck SA, Marousi, Greece) and L-tartaric acid and DL-lactic acid from Panreac (Panreac Química SLU, Barcelona, Spain).

### 2.2. Laboratory-Scale Cross-Flow Filtration Pilot Unit 

All cross-flow filtration experiments were conducted in a custom-made laboratory-scale cross-flow filtration pilot unit, which is depicted in Appendix A. A detailed description of the pilot unit can be found in Kontogiannopoulos et al. [22]. The authors tested two different membrane filtration configurations and several types of membranes for the separation of polyphenols and polysaccharides from wine lees, and concluded that cross-flow configuration had a satisfactory efficiency. In this study, the tubular membranes FPA03 (apparent retention 75% CaCl_2_) and AFC30 (Molecular weight cutoff 200 Daltons, apparent retention 90% CaCl_2_) from PCI Membranes (Hampshire, UK) were used due to their potential for higher fouling resistance. Their characteristics are presented in Table 2. 

The first step in the experimental procedure was to remove the chemical preservatives by rinsing the membranes with DI water. The module was operated in concentration mode for at least 4 h to ensure that both the conductivity and total organic carbon of the feed and the permeate differed by a maximum of 5%. The initial permeability of the clean membranes was calculated by performing clean water flux (CWF) tests at various operating transmembrane pressures (TMP). All runs were performed under ambient conditions. Therefore, the temperature of the feed fluctuated between 20 °C and 30 °C and was recorded constantly. All measurements were corrected to a reference temperature of 25 °C, according to the following equation:(1)$$L_{25\;{}^{\circ}C}=\frac{L_{T\;}\times \eta_{Τ}}{\eta_{25\;{}^{\circ}C}},$$
where *L*_25_
_°C_ = the membrane permeability at the reference temperature of 25 °C (L/m^2^ h bar); *L_T_* = the membrane permeability at the given temperature T (L/m^2^ h bar); *η_T_* = the permeate viscosity at the given temperature T (Pa s); and *η*_25_
_°C_ = the permeate viscosity at the reference temperature of 25 °C (Pa s).

Then, the TMP was increased to a value at least 20% higher than that in the experiments to compact the membranes in recirculation mode for 2 h using DI water. This step was employed to minimize errors in the permeate flux calculations due to membrane compaction and alterations in the initial clean membrane filtration resistance. The cross-flow velocity used in all the compaction steps and the cross-flow filtration experiments was 0.34 m/s, which corresponds to a feed volumetric rate of approximately 2 L/min. The supernatant, after the centrifugation of the wine lees, was tested in concentration mode to determine the variation of permeate flux as a function of the operation time and volumetric concentration factor (*VCF*). The *VCF* was defined as follows:(2)$$VCF=\frac{V_{feed}}{V_{feed}-V_{permeate}}.$$

Samples from all process streams (feed, permeate, and concentrate) were collected at the end of each filtration step and analyzed for pH, conductivity, organic acid concentration (tartaric, malic, lactic, and citric acid), and total and individual polyphenols (quercetin, epicatechin, gallic acid, caffeic acid, and ferulic acid). Furthermore, the separation factor of individual organic acids and/or total organic acids (compound A) and total polyphenols (compound B) was calculated based on the following equation:(3)$$S_{F,\;AB}=\frac{\left(\frac{C_{A}}{C_{B}}\right)permeate}{\left(\frac{C_{A}}{C_{B}}\right)concentrate}.$$

Membranes were cleaned after each experiment according to the following protocol: First, they were thoroughly rinsed with DI water followed by chemical cleaning in recirculation mode. The cleaning solution consisted of DI water that contained 0.5% *v*/*v* Mucasol^®^ (Schülke & Mayr GmbH, Norderstedt, Germany.), as a universal detergent, and 0.15% *v*/*v* NaOCl solution (14% *w*/*v* available Cl), which resulted in approximately 200 mg/L Cl. The pH of the cleaning solution was approximately 10.5. The chemical cleaning lasted for approximately 8 h. 

### 2.3. Spray-Drying of AFC30 Concentrate

The concentrate from nanofiltration was further processed to a dry powder by spray-drying (Buchi Mini Spray Dryer Model 191, Buchi Labortechnik AG, Flawil, Switzerland). Initially, the concentrate was partly evaporated (rotary evaporator Heidolf, Laborota 4003, Heidolph Instruments GMBh & CO KG, Schwabach, Germany) to form a solution of 10% TDS. Preliminary experiments showed that direct drying of the concentrate was not possible because particles were sticking to the drying chamber. Therefore, maltodextrin DE_10_ was added to the solution at a final concentration of 13% (*w*/*v*) as a drying agent. The main constraint in choosing the operating conditions in this case was the outlet temperature of the product, which should be kept as low as possible to avoid degradation of the heat-sensitive polyphenols. Based on preliminary test runs, the chosen operating conditions were: drying air temperature, 140 °C; compressed air flow 0.7 m^3^/h; sample feed rate, 0.82 ± 0.05 g/min; and drying air rate, 21 m^3^/h. The collected powder was sealed in a vacuum package and stored at −20 °C until analysis for its antioxidant properties.

### 2.4. Preparation of Food System for Testing Antioxidant Potential of Dried Concentrates

The antioxidant potential of the spray-dried nanofiltration concentrates was compared against a known synthetic antioxidant (BHT) in a real food system. Fresh minced bovine meat (*M. longissimus dorsi* trimmed of visible fat) was purchased from a local store. It was split into 40-g portions, and the following samples were prepared: Blank (40 g mince + 10 mL DI water); samples with 100, 500, and 1000 mg/kg dried concentrate dissolved in 10 mL DI water; one sample with 100 mg/kg BHT dissolved in 8 mL DI water and 2 mL ethanol; and one sample with 1000 mg/kg maltodextrin dissolved in 10 mL DI water. The sample with maltodextrin was prepared to evaluate maltodextrin’s potential contribution to the antioxidant behavior of the dried concentrates. 

Samples were thoroughly mixed with a stainless-steel spatula while being kept on ice and evenly spread on the flat surface of glass flasks, wrapped in aluminum foil, and stored at 4 °C. Lipid oxidation was followed daily by TBARS and the determination of the oxy-myoglobin (oxyMB) concentration over an eight-day period. Three independent experiments were conducted.

### 2.5. Analytical Methods

#### 2.5.1. HPLC Determination of Organic Acids and Polyphenols

Organic acids were determined by reverse-phase HPLC using a Shimadzu LC-10 AD VP (Kyoto, Japan) liquid chromatograph. Separation was achieved with a AQUASIL C18 (5 μm, 250 mm × 4.6 mm) (Thermo Scientific, Boston, MA, USA) column thermostated at 45 °C, coupled with a Diode Array Detector (SPD-M20A) (Shimadzu). Detection was performed at 210 nm as described by Kontogiannopoulos et al. [22]. Quantification was performed by constructing calibration curves for each organic acid separately (malic, tartaric, lactic, and citric acid).

Polyphenols were also determined by reverse-phase HPLC using a Knauer 1200 system (KNAUER Wissenschaftliche Geräte GmbH, Berlin, Germany) fitted with a C18 column (LiCrospher^®^ 100 RP-18, 250 × 4 mm; 5 µm) and coupled to a UV detector recording the signal at 280 nm, 306 nm, and 365 nm. The injection volume was 20 μL and the analysis method was adapted from Romero-Diez et al. [23]. Quantification was performed by constructing calibration curves for each phenol separately (epicatechin, quercetin, ferulic acid, caffeic acid, p-coumaric acid, resveratrol, and gallic acid). Examples of the calibration curves are given in Appendix A. 

#### 2.5.2. Antioxidant Potential Analyses

DPPH inhibition: This method is based on the evaluation of the free-radical scavenging capacity. The DPPH solution (0.1 g/L in ethanol) was prepared daily, stored in a flask covered with aluminum foil, and kept in the dark at 4 °C between measurements. Radical scavenging activity was determined by measuring the absorbance at 517 nm and was expressed as the inhibition percentage using the equation:% Inhibition DPPH = 100 × [A_DPPH_ − A_sample_]/A_DPPH_, (4)
where A_DPPH_ is the absorbance value of the DPPH blank sample and A_sample_ is the absorbance value of the sample solution.

The FRAP (ferric reducing power) was determined as described by Hayes et al. [24].

#### 2.5.3. Total Phenols

The content of total phenolic compounds, in all samples, was determined using the Folin–Ciocalteu method employing a Helios γ spectrophotometer (Thermo Scientific). Gallic acid was used as a reference standard and the results were expressed as mg gallic acid/L [25].

#### 2.5.4. TBARS

The TBARS were determined according to Vareltzis et al. with small modifications [26]. Samples of 1.5 g in triplicates were carefully taken from the flask and the remaining quantity was spread evenly again on the bottom to cover the whole area and form a layer uniform in height. The sample was added to a test tube containing 5 mL of freshly prepared TCA (7.5% *w*/*v*). The mixture was homogenized with an IKA Tissue Homogenizer at 3000 rpm for 1 min while being kept on ice (Ultra Turrax, IKA-Werke GmbH, Staufen, Germany), vortexed, and centrifuged for 25 min at 2900× *g*. Two milliliters of the supernatant were mixed with 2 mL of TBA solution (0.02 M). The mixture was heated in a water bath for 40 min at a constant temperature of 100 °C. Finally, after cooling down under running tap water, the absorbance was measured spectroscopically at 532 nm. Quantification was achieved by constructing a standard curve with TEP solutions and the results were expressed as MDAeq (μmol/kg sample).

#### 2.5.5. Oxymyoglobin Content

Oxymyoglobin (oxyMB) is responsible for the bright red color of meat. The discoloration of meat is often associated with lipid oxidation and is attributed to the oxidation of oxyMb and myoglobin to form metmyoglobin (metMb) [27]. Spectroscopic determination of oxyMb is achieved by extracting the myoglobin forms from the matrix and measuring the absorbance in different wavelengths. These absorbance values are then used in the following equations to calculate the relative abundance of the three main forms of myoglobin [28]:%metMb = (−0.159 × R_1_ − 0.085 × R_2_ + 1.262 × R_3_ − 0.52) × 100 (5)
%deoMb = (−0.543 × R_1_ + 1.594 × R_2_ + 0.552 × R_3_ − 1.329) × 100 (6)
%oxyMb = (0.722 × R_1_ − 1.432 × R_2_ − 1.659 × R_3_ + 2.599) × 100,(7)
where R_1_ = A_582_/A_525_, R_2_ = A_557_/A_525_, R_3_ = A_503_/A_525_.

Briefly, samples in triplicate (*n* = 3) were homogenized in 10× volume phosphate buffer (0.04 M, pH 6.8) and centrifuged for 30 min at 2900× *g*. Aliquots of the supernatant were directly measured at the different wavelengths in Equations (5)–(7), using phosphate buffer as a blank.

### 2.6. Statistical Analysis

ANOVA (one-way analysis of variance) with Tukey’s test was used to compare means. Significance was reported at the *p* < 0.05 level. Data are presented as mean values ± standard deviation (SD) obtained from three independent analyses, unless otherwise noted. Minitab^®^ 21 (Minitab, Ltd., Coventry, UK) software was used for the statistical analysis.

## 3. Results and Discussion

### 3.1. Membrane Filtration

Cross-flow filtration runs were performed in concentration mode for both membrane types at TMP = 2.0 bar and a cross-flow velocity of 0.34 m/s. It should be noted that the permeate of the ultrafiltration process was used as the feed stream for nanofiltration. Table 3 summarizes the characteristics of all streams (feed solution, permeate, and concentrate) from the cross-flow filtration tests. Specifically, the volume, *VCF*, pH, turbidity, and anti-oxidant properties of all streams for each membrane type are reported. 

The geometrical configuration of membranes plays an important role in both their separation efficiencies and their fouling propensities. In the literature, there are reports of favorable separation efficiencies concerning organic acids and polyphenols using various configurations, such as spiral-wound or flat-sheet. In most laboratory-scale experiments, the raw materials (i.e., wine lees) are centrifuged and then diluted with water to decrease the suspended solids of the feed stream. However, these practices are not applicable at the industrial scale, where less energy-intensive processes, like screeing or sedimentation, are preferred. These processes, however, are not as efficient as centrifugation and dilution at decreasing the concentration of suspended solids. Therefore, in this work, we chose tubular geometry, because it is considered more appropriate for applications involving a high concentration of suspended solids in the feed [22,29,30]. 

Concerning membrane performance, both types of filtration membrane led to a reduction in permeation flux. However, the initial flux reduction rate was more pronounced for the FPA03 membrane. Notably, the initial sharp decline in permeate flux of the ultrafiltration membrane was followed by a plateau value of around 4 L/m^2^/h/bar (LMH/bar) after approximately 4 h of operation. This value is 36% of the initial flux, while for the nanofiltration membrane (AFC30) after the same time of operation the corresponding value was approximately 80% (Figure 1a,b). It seems that the fouling layer was partially removed by the applied cross-flow velocity. The sharp decline in the permeate flux of the ultrafiltration membrane can also be attributed to the removal of compounds with a high fouling propensity (e.g., suspended solids, and possibly other macromolecules). This is further indicated by the turbidity measurements. The turbidity was decreased from 150 in the feed to 0.6 in the permeate. On average, 80% of total polyphenols and 60% of total detected acids passed through the ultrafiltration membrane (Table 4). This observation was in accordance with the higher antioxidant ability of the permeate, as expressed by both FRAP and %DPPH inhibition compared to the concentrate stream of ultrafiltration (*p* < 0.05) (Table 3). However, the ultrafiltration membrane was not successful at efficiently separating organic acids from polyphenols, as confirmed by the calculated separation factors for individual and total acids. Separation factors were in the range of 0.27 to 0.54 (Table 5). Low separation factors for organic acids and polyphenols for ultrafiltration membranes have been reported by other researchers in the case of wine lees and other fruit juices [22,31].

On the other hand, nanofiltration concentrate comprised 64% of the initial TPP amount in the feed stream of the process. The separation efficiency of nanofiltration membrane was 81% (amount in concentrate/amount of TPP in nanofiltration feed). Accordingly, the concentrate of the nanofiltration step showed the highest %DPPH inhibition and FRAP values compared to all other process streams (*p* < 0.05). Tao et al. reported an optimized yield of TPPs from wine lees after ultrasound-assisted extraction of around 60% [32]. Kontogiannopoulos et al. reported a 69% recovery in red wine lees’ TPPs in the concentrate of an HFW1000 nanofiltration membrane [22]. Nanofiltration also achieved a significantly higher separation factor for organic acids and polyphenols, which ranged from 5.17 for tartaric acid to 9.18 for DL-lactic acid (Table 5). 

Polyphenols are characterized by their poor water solubility. However, polysaccharides that are also present in wine lees are highly soluble in water. Possible intermolecular interactions between polyphenols and polysaccharides might account for the high recovery of polyphenols. Several researchers have tested a variety of nanofiltration membranes and shown that there is no significant preferential rejection of phenolic compounds over sugars [22,33,34]. Compounds of similar molecular weight present in the centrifuged wine lees prevent an effective membrane fractionation. Therefore, membrane processing is more suitable for the recovery and concentration of bioactive compounds and especially temperature-sensitive compounds, such as polyphenols. The sequential use of ultra- and nanofiltration is a promising mild process for producing, at ambient temperatures, preconcentrated solutions from wine lees. These solutions can be further concentrated via other processes, such as vacuum evaporation, osmotic evaporation, or spray-drying. On the other hand, Giacobbo et al. reported that composite fluoro polymer UF membranes with MWCO of 1 and 10 kDa were successful for the recovery and fractionation of polyphenols and polysaccharides from second racking wine lees [19]. For both membranes, rejections to polysaccharides were higher than 77% and increased linearly by increasing the TMP in the range 3–15 bar. Higher TMP values led to the formation of an additional selective layer on the membrane surface, therefore increasing the rejection coefficients due to concentration polarization and fouling phenomena [35]. Another factor that can affect permeability and rejection factors besides TMP is temperature. Lopez-Borrel et al. showed that an increase in the operating temperature of nanofiltration membranes led to a significant increase in the permeate flux [21].

The detected organic acids showed a similar pattern in behavior regarding separation at both the ultra- and nanofiltration steps. On the other hand, detected polyphenols exhibited different rejection rates in the nanofiltration process: 62.5% of ferulic acid in the nanofiltration process feed was retained in the concentrate, with 75% of epicatechin and only 47% of quercetin. These differences can be attributed to the structure and chemical properties of each polyphenol. Research has identified that the key factors controlling the efficiency of a membrane separation process are the hydrophobic and polar polyphenol–membrane surface interactions, as hydrogen bonding, in addition to the steric hindrance [18]. 

### 3.2. Spray-Drying of Nanofiltration Condensate

Spray-drying is an appealing drying technique in the food industry due to its availability, versatility, productivity, and reduced operational costs [36]. Liquid extracts of polyphenols from various sources can be spray-dried and produce microencapsulated powders with high polyphenolic content [13,37]. However, phenolic compounds pose a challenge to the industry, because they are sensitive to various processing and/or storage conditions, such as light, pH, high temperature, and radiation. Therefore, special care needs to be taken when processing and handling polyphenols in order to retain their biological activities [38]. Encapsulation, among other functionalities, protects heat-sensitive bio-active compounds, such as polyphenols, during drying. Furthermore, during preliminary experiments it was noticed that direct drying of the concentrate led to the sticking of the particles on the drying chamber wall. Stickiness problems in spray-drying are related to the glass transition temperature (Tg) of the material being dried. When drying takes place around the Tg, the particles of powder are in the glass transition state, also known as plastic or rubbery state, and stick to the surfaces of the equipment they come in contact with [39]. Therefore, there was a need for a drying agent to encapsulate the polyphenols and increase the glass transition temperature. Maltodextrin (MD) was chosen for its several advantages. MD is a widespread biopolymer used as the carrier for polyphenolic encapsulation because it exhibits low viscosity at high solid contents and has good solubility, notable heat protection capacity, long-term resistance, and a pleasant flavor [12,40]. 

The operating parameters were adapted from the literature, after preliminary experiments, with the aim of making the outlet temperature of the dried powder as low as possible to avoid polyphenol degradation. Under the chosen conditions, the outlet temperature of the dry powder was 70 °C and the solid yield (weight of powder/initial total solids) was 52%, comparable to that for similar processes reported in literature [12]. The powder contained 857.35 ppm polyphenols, retaining 64% of the phenolic content of the nanofiltration concentrate. It is probable that the limited exposure to high temperature during drying, as well as the use of MD as a drying agent, protected the thermosensitive polyphenols from degradation during spray-drying. The optimization of this process, employing a multivariate approach, would probably lead to an improved yield of polyphenols in the powder. Spray drying did not significantly affect the antioxidant properties (%DPPH inhibition was 89.73% and FRAP 33.4 μmol TPTZ/L) of the powder compared to the properties of the concentrate prior to drying (*p* > 0.05, Table 3). High antioxidant activities were also observed in spray-dried wine wastes, when alginate microbeads were used [13], accompanied with high retention of the initial phenol content or anthocyanins [14]. 

### 3.3. Antioxidant Potential of Spray-Dried Nanofiltration Powder

Lipid oxidation is one of the major causes of food quality deterioration. In muscle foods, oxidation results in the development of off flavors (secondary oxidation products), characteristic of rancidity, and eventually renders the product unacceptable to consumers, decreasing its shelf life [41]. Quality deterioration also includes color changes, textural modifications, formation of low molecular volatile compounds, and a general loss of nutritional quality, due to the degradation of antioxidants and vitamins, as well as polyunsaturated fatty acids that serve as a substrate for lipid oxidation [42].

Besides in vitro testing of the anti-oxidant potential of a bioactive substance, it is important to test it in food systems more similar to real foods. Lipid oxidation is a complex mechanism that can involve free radicals, inorganic polyvalent cations, oxygen, organic iron (i.e., heme-iron), and enzymes and can be affected by the type, physical state, and concentration of lipids and other food constituents, such as proteins. In contrast to in vitro systems, real foods contain a variety of endogenous antioxidants, with both enzymatic (e.g., catalase, superoxide dismutase) and nonenzymatic components (e.g., ascorbic acid, tocopherols, carotenoids) [43]. Furthermore, in the case of muscle foods, heme and/or myoglobin mediated lipid oxidation has been shown to be one of the prevalent mechanisms for lipid oxidation, since hemoglobin and myoglobin act as potent pro-oxidants [44,45]. 

The spray-dried concentrate of the nanofiltration step was able to significantly retard the lipid oxidation of minced beef samples at concentrations of 500 ppm and 1000 ppm compared to the control (Figure 2). This protection against lipid oxidation can be attributed to its polyphenolic content. Maltodextrin, used as a drying agent to encapsulate the concentrate, did not have any significant effect on the oxidative stability of the samples. However, the synthetic antioxidant BHT, at the maximum allowable concentration of 100 ppm, offered better protection against lipid oxidation than any sample with the spray-dried concentrate. 

Oxymyoglobin (oxyMB) is responsible for the bright red color of meat. Discoloration of meat is often associated with lipid oxidation and is attributed to the oxidation of oxyMb and myoglobin to form metmyoglobin (metMb) [27]. Spectroscopic determination of oxyMb is achieved by extracting the myoglobin forms from the matrix and measuring the absorbance in different wavelengths. By applying Equations (5)–(7), we were able to determine the oxyMb content of the samples during an eight-day refrigerated storage period. It was observed (Figure 3) that the oxymyoglobin content in the sample with 1000 ppm dried concentrate powder was higher than that in the control throughout the storage period. It was also higher than the sample with the synthetic BHT antioxidant until the second day of storage. From the third day of storage until the end of the experiment (day 8), there was no significant difference between these samples. This is an indication that the spray-dried concentrate offers better initial protection against myoglobin-mediated lipid oxidation than BHT. Maltodextrin did not seem to play any role in oxyMb protection, as was the case with lipid oxidation.

## 4. Conclusions

This work showed that integrating the sequential ultra- and nanofiltration membrane process with spray-drying of the final concentrate is a promising technology to recover and utilize polyphenols from wine lees with industrial applications. A more detailed investigation of the operating parameters of the membranes might lead to improved separation of the organic acids and polyphenols. It was also shown that spray-drying of this concentrate is possible, as long as a drying agent is used to avoid the dried powder sticking to the drying chamber wall. The resulting powder retained most of its polyphenol content, along with the antioxidant properties, and was successfully applied to a real food system to retard lipid oxidation. Therefore, membrane processing is an attractive alternative process for producing, at ambient temperatures, concentrated solutions of polyphenolic substances with antioxidant properties from wine lees.

## Figures and Tables

**Figure 1 membranes-12-00353-f001:**
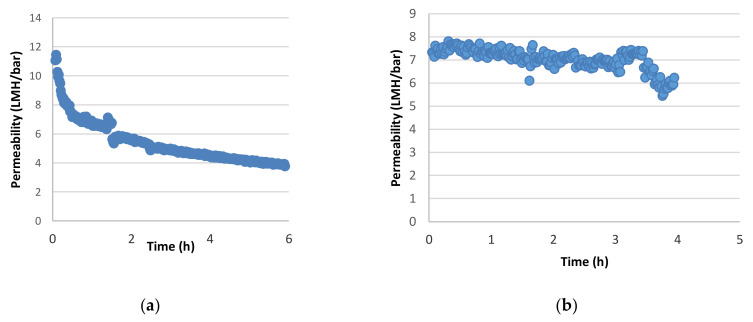
Permeability of (**a**) FPA03 (ultrafiltration) and (**b**) AFC30 (nanofiltration).

**Figure 2 membranes-12-00353-f002:**
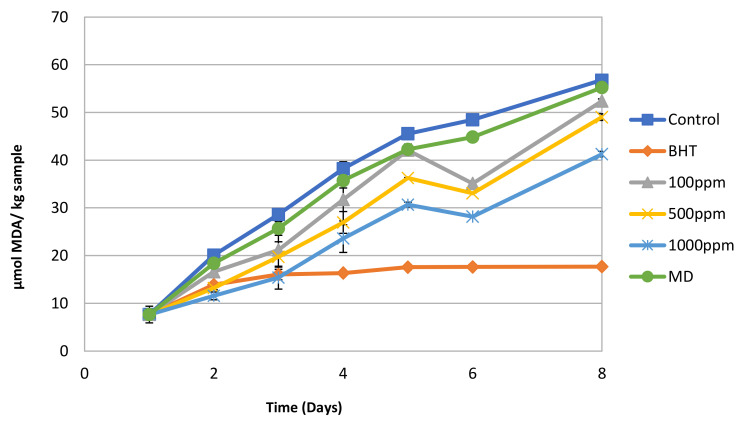
TBARS of bovine mince samples during refrigerated (4 °C) storage. All samples were comprised of 40 g mince and 10 mL DI water, in which the corresponding amount of the spray-dried concentrate was dissolved. The BHT sample contained 100 ppm BHT dissolved in 8 mL DI water and 2 mL ethanol. The control consisted only of 40 g mince and 10 mL DI water.

**Figure 3 membranes-12-00353-f003:**
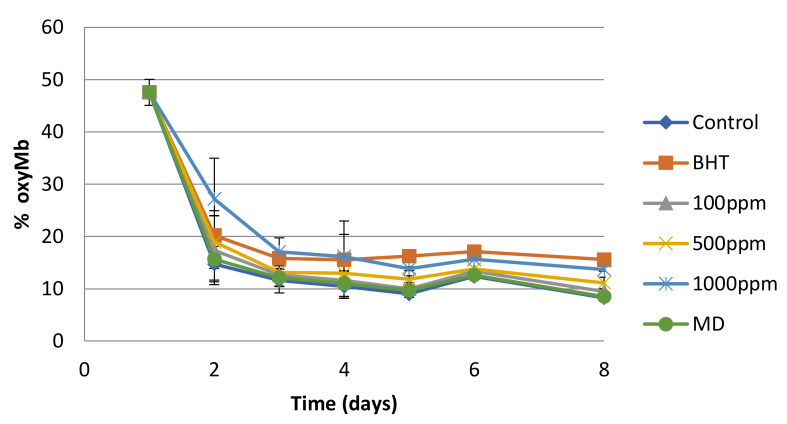
%oxyMyoglobin during refrigerated storage of minced bovine meat.

**Table 1 membranes-12-00353-t001:** Basic physicochemical properties of collected wine lees ^1^.

Property	Value
% moisture	95.6
Total solids (g/L)	68.5
Total dissolved solids (TDS) (g/L)	23.7
pH	4.09
Conductivity (mS/cm)	2.06

^1^ Values are reported as mean of three independent measurements (*n* = 3).

**Table 2 membranes-12-00353-t002:** Main characteristics of filtration membranes.

Membrane	Application	Material	pH Range	Max. Operating Pressure (bar)	Max. Temperature (°C)
FPA03	Ultrafiltration (UF)	PVDF	1.5–10.5	7	60
AFC30	Nanofiltration (NF)	Polyamide, TFC	1.5–9.5	60	60

**Table 3 membranes-12-00353-t003:** Properties and phenolic content of process streams and spray dried NF concentrate (average ± SD, *n* = 3 *).

Membrane Type	Stream	Volume (L)	*VCF*	pH	Turbidity (NTU)	%DPPH Inhibition	FRAP (μmol TPTZ/L)	Total Phenols (mg gallic acid/L)
UF(FPA03)	Feed	10		4.09	150	61.52 ± 0.097 ^a^	15.2 ± 0.078 ^a^	315.27 ± 0.21 ^a^
	Permeate	6.2	2.6	4.86	0.6	83.48 ± 0.066 ^b^	22.2 ± 0.093 ^b^	403.46 ± 0.13 ^b^
	Concentrate	3.8		4.07	206	53.53 ± 0.117 ^c^	13.9 ± 0.115 ^a^	138.33 ± 0.10 ^c^
NF(AFC30)	Feed	6.2		4.86	0.6	83.48 ± 0.066 ^a^	22.2 ± 0.093 ^b^	403.46 ± 0.153 ^b^
	Permeate	4.7	4.1	4.05	0.05	18.19 ± 0.058 ^d^	3.78 ± 0.034 ^c^	58.74 ± 0. 111 ^d^
	Concentrate	1.5		3.9	0.5	92.85 ± 0.070 ^e^	36.3 ± 0.119 ^d^	1351.15 ± 0.063 ^e^

* Different superscripts (a–e) correspond to significant differences, *p* < 0.05.

**Table 4 membranes-12-00353-t004:** Concentrations of detected acids and phenols in process streams (average, *n* = 2).

Substance (ppm)	Feed	Permeate UF (FPA03)	Concentrate UF (FPA03)	Permeate NF (AFC30)	Concentrate NF (AFC30)
Tartaric acid	1630	1660	1060	820	3645
DL-malic acid	420	455	335	310	800
DL-lactic acid	1520	1240	1560	795	1550
Citric acid	410	460	300	320	840
Total acids (mg)	39,800	23,653	12,369	10,551.5	10,252.5
Ferulic acid	51	60	32	23	155
Epicatechin	100	116	43	34	360
Quercetin	17.5	19.5	13	15	38
Total phenols (mg)	1685	1212.1	334.4	338.4	829.5

**Table 5 membranes-12-00353-t005:** Separation factors (average, *n* = 3).

Compound	Separation Factors (SF_AB_)	
	UF (FPA03)	NF (AFC30)
L-tartaric acid	0.54	5.17
DL- malic acid	0.47	8.95
DL- lactic acid	0.27	9.18
Citric acid	0.53	8.79
Total acids	0.40	5.76

SF_AB_ = [C_A_/C_B_] permeate/[C_A_/C_B_] concentrate, where A is each individual acid or the total acids and B is the total polyphenols.

## Data Availability

Not applicable.

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
