# Peer review of "Sequential Membrane Filtration to Recover Polyphenols and Organic Acids from Red Wine Lees: The Antioxidant Properties of the Spray-Dried Concentrate"

_membranes, 2022, doi:10.3390/membranes12040353_

Round 1
Reviewer 1 Report
Integrated UF/NF/Spray drying process was employed to produce polyphenol powder from wine lees. This approach is in the line of bio-circular-green economy to promote sustainable development with fully utilize bio resources . The result is helpful to winery but also to related industry. The manuscript was well organized and presented. It should be published as it is.
Reviewer 2 Report
The author designed ultra- and nano-membrane filtration to recover and utilize polyphenols from wine lees. The study is interesting and has good applications. In my point, there extists some concerns as below:
1. check the whole manuscript especially tables, numbers should be expressed consistently (value in table 1 is not the same as ph range in table 2). Also, the number of decimal points should be the same (check the whole manuscript).
2. Line 211 and 230, 4000 rpm should be changed as XXXX g.
3. The format of table 3 is confused.
4. In table 4, why the number of sample is 2?It was not statistically significant. Besides, in the feed, the total phenols is 1685, but the sum of ferulic acid, epicatechin and quercetin is 168.5, so what are the other phenolic componentsthen?
Reviewer 3 Report
This paper deals with the membrane separation of process polyphenols and organic acids. Reviewer understands the importance of the recovery of polyphenols and organic acids. However, this paper seems a technical paper and does not include a novelty. There is no academic discussion. Authors do not add future prospective of the presented research. Authors describe only experimental results. So the paper is rejected.
Reviewer 4 Report
The manuscript is an interesting study on the use of combined membrane separation (i.e. UF followed by NF) for the separation and concentration of polyphenolic substances from red wine lees. The manuscript’s subject and objectives are within the journal’s aims and content and the presented work (to the best of the reviewer’s knowledge) is original. The novelty of the manuscript’s content is moderate, however, it is one of the few studies that results to a final powdered product, which is evaluated in real food applications. The quality of the presentation and the English language is high and minor issues should be addressed by the authors:
- Lines 36-38: Please consider rephrasing the sentence as follows: "It has been estimated that 0.13 t of marc, 0.06 t of lees, 0.03 t of 37 bunches, and 1.65 m3 of wastewater are generated for each ton of processed grapes [5]."
- Line 41: Please add commas as follows: "Wine lees, as the second most generated byproduct of vinification process, have not..."
- Line 49: ethanol and acetone cannot be characterised as considerably toxic, maybe the phrase "toxic and/or irritating" is more precise.
- Line 58: Please consider rephrasing the sentence as follows: "...to produce a concentrated solution, rich in polyphenols, with..."
- Line 59: Please consider removing the phrase "...conducted by members of our team..."
- Line 77: Please replace "to" with "into".
- Line 98: Please consider replacing "Here," with "In this study,".
- Line 108: Please add a space between "clean" and "water".
- Line 125: Please consider adding commas as follows: "The supernatant, after the centrifugation of the wine lees, was tested in...".
- Line 129: Please specify the time point at which the analysed samples were collected. Was it at the end of the membrane filtration experiments?
- Line 136: Please consider replacing "...in the following manner:" with "...according to the following protocol:".
- Line 187: Given that both DPPH inhibition and FRAP were used to characterise the antioxidant capacity, maybe the authors could change the title to "Antioxidant activity analyses"?
- Line 221: Please replace "measure" with "measuring".
- Line 242: Can the authors confirm that the TMP used in both membrane processes was the same? TMP of 2 bar is rather low for NF processes.
- Table 3 needs to be formatted to be improved concerning its presentation quality.
- Line 251: The geometrical configuration also affects the fouling behavior of membrane processes. Maybe the authors should rephrase the sentence as follows: "...play an important role both in their separation efficiencies and their fouling propensities.".
- Line 254: Please consider deleting the word "though".
- Lines 268-269: The main conclusion of these results is that UF membrane seems to successfully remove compounds (e.g. suspended solids, and possibly macromolecules) with high fouling propensity. Do the authors agree?
- Lines 293, 304 and elsewhere: The authors use the terms "nanomembrane filtration" or "nanomembranes" which are incorrect. Please replace these terms with "nanofiltration" or "nanofiltration membrane" or "nanofilitration process".
- Line 338: Please consider rephrasing the sentence as follows: "Spray drying is an appealing drying technique to the food industry due...".
- Lines 365-367: Can the authors specify which exact results support these conclusion/statement?
- Line 407: Please replace "measure" with "measuring".
- Lines 411-412: The sentence "...and similar for the rest of the storage duration." is unclear; can the authors detail this statement?
- Line 444: Please consider rephrasing the sentence as follows: "...concentrated solutions of polyphenolic substances with antioxidant properties from wine lees.".
- Supplementary Materials: Given the rather low number of Figures and Tables in the manuscript, Table S1 could be incorporated in the manuscript's text.
Round 2
Reviewer 2 Report
I have no comments this time
Author Response
We would like to thank the reviewer for the comments and the final recommendation.
Reviewer 3 Report
Unfortunately, the paper is still not including a novelty. Nano filtration Membrane System for the Recovery of Polyphenols from Wine Lees (10.3390/membranes12020240) was studied before by other researchers and “spray drying” is a method which applied frequently.
In the conclusion section, the authors mentioned that “A more detailed investigation on the operating parameters of the membranes might lead to improved performance regarding separating organic acids and polyphenols” if they decided to this why do they suggest that “membrane is an attractive alternative method” in Lines 468-470 if they need more detailed studies for improving performance of separation.
